# The importance of intermediate filaments in the shape maintenance of myoblast model tissues

Irène Nagle[1], Florence Delort[2], Sylvie Hénon[1], Claire Wilhelm[1†], Sabrina Batonnet-Pichon[2], Myriam Reffay[1]*

[1]Laboratoire Matière et Systèmes Complexes, UMR 7057, Université Paris Cité and CNRS, Paris, France; [2]Laboratoire Biologie Fonctionnelle et Adaptative, UMR 8251, Université Paris Cité and CNRS, Paris, France

*For correspondence:
myriam.reffay@u-paris.fr

Present address: [†]Laboratoire Physico-Chimie Curie, UMR 168, CNRS, Institut Curie, Université PSL, Sorbonne Universite, Paris, France

Competing interest: The authors declare that no competing interests exist.

**Abstract** Liquid and elastic behaviours of tissues drive their morphology and response to the environment. They appear as the first insight into tissue mechanics. We explore the role of individual cell properties on spheroids of mouse muscle precursor cells and investigate the role of intermediate filaments on surface tension and Young's modulus. By flattening multicellular myoblast aggregates under magnetic constraint, we measure their rigidity and surface tension and show that they act as highly sensitive macroscopic reporters closely related to microscopic local tension and effective adhesion. Shedding light on the major contributions of acto-myosin contractility, actin organization, and intercellular adhesions, we reveal the role of a major component of intermediate filaments in the muscle, desmin and its organization, on the macroscopic mechanics of these tissue models. Implicated in the mechanical and shape integrity of cells, intermediate filaments are found to be crucial to the mechanics of unorganized muscle tissue models even at an early stage of differentiation both in terms of elasticity and surface tension.

## Editor's evaluation

This important work studied the determinants of key physical properties of multicellular assemblies using magnetic flattening of spheroids. The key and convincing result is that intermediate filaments could also be implicated in the setting of the elastic properties of these assemblies, shedding light on this central cellular component and how their modifications could be important to the understanding of some pathologies.

## Introduction

Tissue-forming cells interact with each other and with their environment (*Barone and Heisenberg, 2012*; *Humphrey et al., 2014*; *Indana and Chaudhuri, 2021*) giving rise to interesting viscoelastic fluid behaviours (*Lecuit and Lenne, 2007*) that are determinant both in epithelia (*Mongera et al., 2018*) and 3D tissue-forming systems (*Maître et al., 2012*). Physical properties such as surface tension (*Ehrig et al., 2019*; *Harmand et al., 2021*) and viscosity (*Stirbat et al., 2013*) can be introduced to predict tissue organization and shape (*Etournay et al., 2015*). Global mechanical characteristics of tissues emerge from individual cell components and their interplay (*Jakab et al., 2008*; *Dolega et al., 2021*; *Grosser et al., 2021*), but whether they could be good reporters of individual cell behaviour and global organization is still unknown. Multicellular aggregates (*Kalantarian et al., 2009*; *Stirbat et al., 2013*) prove to be powerful tools to apprehend fundamental biological processes as morphogenesis (*Costa et al., 2016*), development (*Birey et al., 2017*), and tumorigenesis (*Montel et al.,*

*2011*; *Gunti et al., 2021*). They are able to mimic various biological phenomena (*Nikolaev et al., 2020*), and as model systems, are easier to use and monitor than in vivo tissues (*Bassi et al., 2021*). Combining simplicity, reproducibility, and biological significance, they are a system of choice both for biophysics and computation (*Gonzalez-Rodriguez et al., 2012*; *Martin and Risler, 2021*; *Ackermann et al., 2021*). Myoblasts are widely studied cells to understand myogenesis because of their interest in myopathy modelling (*Smoak et al., 2019*) and drug testing (*Zhuang et al., 2020*). Muscle cell mutations implicated in numerous diseases have been extensively studied starting from symptoms to molecular origins identification (*Batonnet-Pichon et al., 2017*; *Jungbluth et al., 2018*). Among these mutations, the ones concerning intermediate filaments are very important (*Dutour-Provenzano and Etienne-Manneville, 2021*). Intermediate filament network is essential in muscle development, provides mechanical integrity to the cell (*Chang and Goldman, 2004*), and plays a major role in the dynamic response to mechanical stimulation (*Charrier et al., 2016*). Desmin, as a major component of intermediate filaments specifically expressed in smooth, skeletal, and cardiac muscles, presents some mutations associated to muscle defects and myopathies. However, the early effects of these mutations in tissues are unclear due to a lack of in vitro biomimetic muscle systems (*Hofemeier et al., 2021*). To address these limitations, we focus on mouse myoblast cells (C2C12) to test the sensitivity of 3D unorganized early-stage muscle tissue models to individual cell modifications. C2C12 are adhesive and highly contractile cells (unsurprisingly regarding their function). Their high assumed surface tension makes them challenging to characterize. We designed an integrated sensitive magnetic tensiometer (*Mazuel et al., 2015*) to form stimulable myoblast-derived tissues and measure their surface tension and elasticity. In this study, we explore the interplay between macroscopic properties of model muscle tissues and the molecular or cellular processes. We investigate how surface tension and Young's modulus represent appealing tools to determine from the tissue scale the microscopic properties. While actin and cadherins are implicated in the surface tension of multicellular aggregates or embryos (*Foty and Steinberg, 2005*), the role of intermediate filaments in tissue shape maintenance has never been identified. Mutations in intermediate filaments severely hinder individual cell nanomechanical properties (*Herrmann et al., 2007*), and their network supports the shape of individual cells (*Goldman et al., 1996*) and withstands applied constraints. Looking at the interplay between their organization and tissue shape maintenance or the tissue elasticity is thus primordial, especially in the context of muscle. We look at cells expressing mutated desmin and exhibiting organization defects of the desmin cellular network to shed light on the crucial role of intermediate filaments on muscle tissue model global mechanics at an early stage of differentiation.

## Results

### Magnetic tensiometer for multicellular aggregates

C2C12 cells are labelled with superparamagnetic nanoparticles without altering their biological capacities (*Van de Walle et al., 2020a*; *Van de Walle et al., 2020b*) nor inducing hypoxia or apoptosis (*Figure 1—figure supplements 1 and 2*). It is then possible to organize and stimulate them at will using external magnets. Magnets first drive cells in agarose moulds to create spheroids of controlled size (*Mazuel et al., 2015*) and content as inhibitors or reagents can be added at this stage (*Figure 1a*). Cohesive spheroids are obtained within 12 hr (*Figure 1b*) and their side profile is imaged (*Figure 1c*). Multicellular spheroids are fulfilled with cohesive cells that organize to form a tissue model with rounded cells at the core of the aggregate having cortical actin and more elongated cells at the periphery having contractile actin network. Their profile is recorded while a magnet is approached (*Figure 1—figure supplements 3 and 4*). Assuming that a multicellular cohesive aggregate can be modelled as a continuous elastic medium (as supported by confocal imaging, *Figure 1b*), surface tension and magnetic forces in volume compete to determine the equilibrium shape (*Kalantarian et al., 2009*) while Young's modulus is at stake for the contact area (*Mazuel et al., 2015*; *Figure 1c*, *Figure 1—figure supplement 3*). The height, the width of the aggregate and the ending points of the contact zone are pointed in the Tensio𝕏 application (*Figure 1—figure supplement 3*), a dedicated MATLAB interface we developed to fit the obtained profile. The elasticity and the capillary parameter $c = \frac{M_v \ grad(B)}{\gamma}$ ($M_v$, $\gamma$, and $B$ denote the magnetic moment per unit of volume, the spheroid surface tension, and the magnetic field, respectively) (*Kalantarian et al., 2009*) are extracted. By knowing the magnetic volume moment, the tissue surface tension is easily deduced from $c$. The error measurement

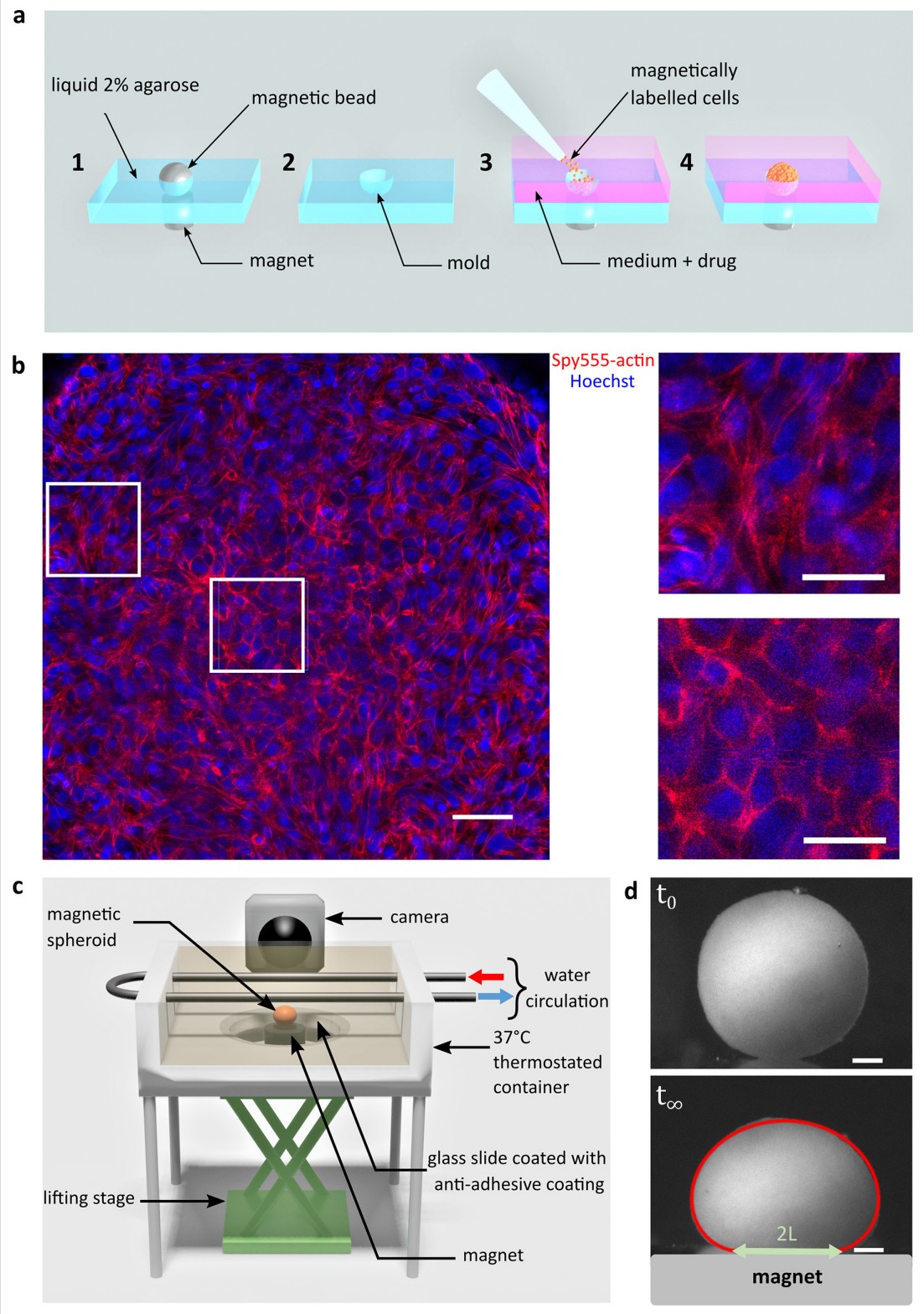

**Figure 1.** Magnetic tensiometer integrated measurement set-up. (**a**) Schematic of the magnetic moulding process. A network of calibrated size steel beads deposited over cylindrical magnets is embedded in heated 2% liquid agarose (1). After agarose gelling, beads are removed creating a semi-spherical mould (2). Magnetically labelled cells are seeded in these non-adhesive-treated moulds. Magnets placed below each mould drive the cells inside the moulds (3). Due to high local cell density, cell–cell contacts develop to form cohesive multicellular aggregates within 12 hr (4). (**b**) Left panel: confocal image of a multicellular aggregate at 150 μm from the top of the aggregate. Actin (red) and nuclei (blue) are labelled. Scale bar: 50 μm. Right

*Figure 1 continued on next page*

*Figure 1 continued*

panels: 75-µm-long zooms for cells at the interface (up) or in the centre of the multicellular aggregate (down). The organization of actin depends on the location of the cell within the aggregate. While it is mainly cortical for cells in the centre, actin is contractile at the periphery of the aggregate to maintain its shape. Scale bar: 25 µm. (**c**) Schematic of the magnetic force tensiometer set-up. A temperature-regulated tank (37°C) is sealed with a non-adhesive glass slide to ensure non-wetting conditions for the multicellular spheroid formed by magnetic moulding ('Methods' and **a**). A cylindrical neodymium permanent magnet (6 mm diameter and height, 530 mT, $\mathrm{grad(B)} = 170\,\mathrm{T.m^{-1}}$) is positioned underneath and approached with respect to a lifting stage. The tank is filled with a transparent culture medium, and the aggregate side profile is monitored with a camera. (**d**) Representative pictures of C2C12 spheroid profiles. Top and bottom pictures show, respectively, a C2C12 spheroid before the magnet approach ($t_0$) and under magnetic flattening when the equilibrium shape is reached ($t_\infty$). The dynamics is shown in *Figure 1—figure supplement 4*. The spheroid surface tension is measured by fitting the aggregate shape with Laplace profile (red line) while the elastic modulus is extracted from the radius of the contact zone $L$ (green arrow) using Hertz theory as described in *Mazuel et al., 2015*; *Figure 1—figure supplement 3*. In this picture, $\gamma = 21\,\mathrm{mN/m}$ and $E = 100\,\mathrm{Pa}$. Scale bar: 200 µm.

The online version of this article includes the following source data and figure supplement(s) for figure 1:

**Figure supplement 1.** Metabolic activity for unlabelled (CTL) and labelled cells ([Fe] = 2–16 mM and 2 hr incubation time), 2 hr (D0), and 1 day (D1) after nanoparticle incorporation.

**Figure supplement 1—source data 1.** Source data of the Alamar-Blue test values for the different magnetic labelling conditions reported in *Figure 1—figure supplement 1*.

**Figure supplement 2.** Immunofluorescence images of C2C12 spheroid cryosections to confirm non-hypoxic and non-apoptotic conditions in spheroids.

**Figure supplement 3.** Extraction of experimental volume, width, and height of the spheroid.

**Figure supplement 4.** Relative height of the aggregate as a function of time for control, latrunculin A (LatA), EGTA, and (±)-blebbistatin ((±)-Bleb) conditions.

**Figure supplement 4—source data 1.** Source data of the evolution of the height of multicellular aggregates presented on *Figure 1—figure supplement 4*.

on the deformations is around 2–5 µm, leading to a precision in the range of 5–20% for surface tension and 5–10% for the elasticity of C2C12 multicellular spheroids. In this regard, it should be noted that the surface tension of C2C12 spheroids is two orders of magnitude greater than that of classically studied cells such as F9 cells, for example. The error increases with surface tension because the spheroids are less deformed, which makes C2C12 cells challenging to characterize. This error is smaller than the inherent distribution measured over different aggregates (*Figure 2a and b*).

## Relation between macroscopic properties and molecular or cellular characteristics: A multi-contribution pattern

Numerous molecular origins of spheroid surface tension are identified. While differential adhesion hypothesis was first evidenced by *Steinberg, 1963* and related to the level of cadherins (*Foty and Steinberg, 2005*), the role of actin cortex was pointed out in individual cells (*Chugh et al., 2017*), spheroids (*Stirbat et al., 2013*), or embryogenesis (*Heer and Martin, 2017*), evidencing a differential interfacial tension hypothesis combining the influence of adhesion and cell surface tension (*Manning et al., 2010*). By comparing the energy of cells in the core of the aggregate to the one at the interface, the surface tension is given by

$$\gamma = T_{CM} - \frac{1}{2}(2T_{CC} - J_{CC}) \tag{1}$$

where $T_{CM}$, $T_{CC}$, and $J_{CC}$ denote the cortical tension at the cell–medium (CM) or at the cell–cell (CC) interface and the intercellular surface adhesion energy ($J_{CC} > 0$), respectively (*Stirbat et al., 2013*). We explore this multi-parameter influence by looking at CC contacts inhibition and changes in actin structure or acto-myosin contractility (*Figure 2*). Intercellular adhesion is mediated by multiple CC adhesion proteins, among which cadherins play a key role. EGTA, as a calcium chelator, reduces the efficiency of this homophilic adhesion and modulates CC adhesion strength. In multicellular aggregates, EGTA has an impact on the acto-myosin network (*Figure 3f*, *Figure 2—figure supplement 1*). By reducing CC contacts, it impairs the formation of the contractile network of acto-myosin at the periphery of the aggregate. At the single-cell level, F-actin becomes mainly cortical whatever the cell location inside the aggregate is. It may be related to a lack of strong enough CC adhesions to maintain actin network and a co-regulation of actin and cadherin tension. EGTA addition leads to a more than fivefold decrease in both Young's modulus and surface tension (*Figure 2a and b*). The

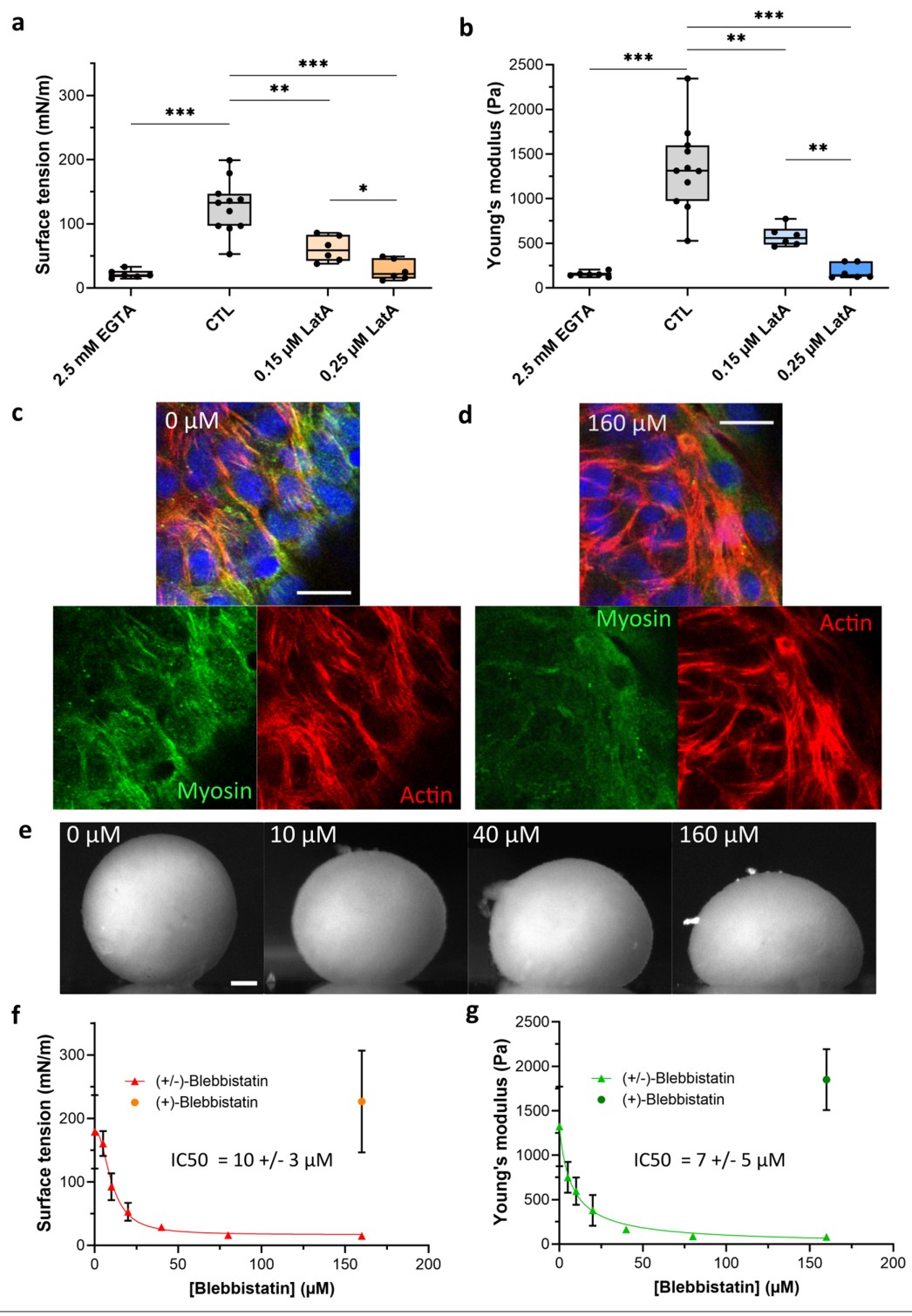

**Figure 2.** Co-action of cortical tension and intercellular adhesions in multicellular spheroid surface tension and Young's modulus. (**a, b**) Variation of surface tension (**a**) and Young's modulus (**b**) of C2C12 spheroids for 2.5 mM EGTA (calcium chelator), 0.15 μM and 0.25 μM latrunculin A (actin disruptor). Floating bars represent min to max variations, and the midline indicates the median. (**c**) Representative confocal images of acto-myosin organization for cells at the periphery of an untreated multicellular aggregate. Nuclei (blue), F-actin (red) and phospho-myosin (green) are labelled. Actin and myosin networks mostly colocalize and form a contractile radial organization. Scale bar: 20 μm. (**d**) Representative confocal image of acto-myosin organization for cells at the

*Figure 2 continued on next page*

*Figure 2 continued*

periphery of a multicellular aggregate treated with 160 µM (±)-blebbistatin. Nuclei (blue), F-actin (red), and phospho-myosin (green) are labelled. Myosin network is almost disrupted compared to actin network. Scale bar: 20 µm. (**e**) Representative pictures of C2C12 spheroids under magnetic flattening at equilibrium for (±)-blebbistatin concentration ranging from 0 to 160 µM are given for comparison. Scale bar: 200 µm. Variations of both surface tension (**f**) and Young's modulus (**g**) with (±)-blebbistatin concentrations are reported. Inhibition curves are fitted for the surface tension (red curve) and the Young's modulus (green curve) with a dose–response providing an $IC_{50}$ value for the (±)-blebbistatin of 10 ± 3 µM and 7 ± 5 µM, respectively, corresponding to an $IC_{50}$ for the (-) active enantiomer of blebbistatin around 6 ± 2 µM and 4 ±3 µM, respectively, as the ratio of the active/negative form is around 50–60%.

The online version of this article includes the following source data and figure supplement(s) for figure 2:

**Source data 1.** Source data of surface tension and Young's modulus measurements for control cell aggregates, EGTA, or latrunculin A-treated cell aggregates reported in *Figure 2a and b*.

**Source data 2.** Source data of the surface tension and Young's modulus measurements for control cell aggregates and blebbistatin-treated cell aggregates reported in *Figure 2f and g*.

**Figure supplement 1.** Immunofluorescence images of C2C12 spheroid cryosections for control conditions (**a**), 0.25 µM latrunculin A (**b**), and 2.5 mM EGTA (**c**).

**Figure supplement 2.** Young's modulus of C2C12 spheroids as a function of their surface tension.

**Figure supplement 3.** Immunofluorescence images of C2C12 spheroid cryosections for 160 µM (+)-blebbistatin (**a**) and 160 µM (±)-blebbistatin (**b**).

---

relationship between adhesion bond energy and tissue surface tension (*Foty and Steinberg, 2005*) is thus tested. An apparent proportionality between surface tension and Young's modulus is obtained (*Figure 2—figure supplement 2*), reminiscent of the one observed between surface tension and elasticity of the whole aggregate (itself related to cortical tension) through co-regulation mechanisms (*Yu et al., 2018*).

Latrunculin A disrupts the actin filaments by binding to actin monomers, thus precluding its polymerization (*Coué et al., 1987*) its addition gives some non-connected patches of actin filaments with a lack of long-range organization (*Figure 3f*, *Figure 2—figure supplement 1*). Young's modulus, as well as surface tension, is dramatically decreased by its presence (*Figure 2a and b*). Besides, its action on macroscopic properties is dose-dependent as seen by the comparison between 0.15 µM (twofold decrease) and 0.25 µM (five- to sevenfold decrease) concentrations. Our results are consistent with the dependence previously noticed between surface tension and viscosity for latrunculin-treated cells (*Jakab et al., 2008*) and extend this dependence to stiffness.

To further test the sensitivity of the magnetic tensiometer, we selected reagents with a wider accessible range still allowing cohesive spheroid formation (*Figure 2—figure supplement 3*). (-)-Blebbistatin inhibits contractility by blocking myosins. (±)-Blebbistatin is a cell-permeable mixed compound containing around 50–60% of the active negative enantiomer that acts as a selective, potent, and reversible inhibitor of non-muscle myosin II without affecting actin filaments assembly. In the multicellular aggregates environment, it impairs the activity of myosins reducing contractility (*Figure 2c and d*). Its inhibition potency is quantified in vitro by $IC_{50}$ values ranging from 2 to 7 µM depending on myosin type (*Zhang et al., 2017*). In our experiments, (±)-blebbistatin is used in a wide range of concentrations (0–160 µM) and leads to an increase in the aggregate deformation with increasing concentration (*Figure 2e*) while the inactive (+)-blebbistatin does not affect mechanical properties. Surface tension and Young's modulus dramatically decrease with (-)-blebbistatin concentration (*Figure 2f and g*) with a decay of up to tenfold. The extracted $IC_{50}$ is around 6 ± 2 µM for surface tension and 4 ±3 µM for elasticity, reproducing the one obtained at the molecular level (as the ratio of the active negative enantiomer is around 50–60%).

## Correlation with geometrical analysis: Tensions at the interfaces

By analogy with fluids, surface tension in multicellular aggregates arises from the energy difference between cells at the interface with the medium and cells surrounded by others. The shape of cells at the surface of multicellular aggregates can thus be used to relate tissue surface tension to cell tensions. Looking at cell surface morphology on multicellular aggregate cryosections (*Figure 3a*) upon the use of the different drugs, we are able to test the relation between surface cell shape and arrangement,

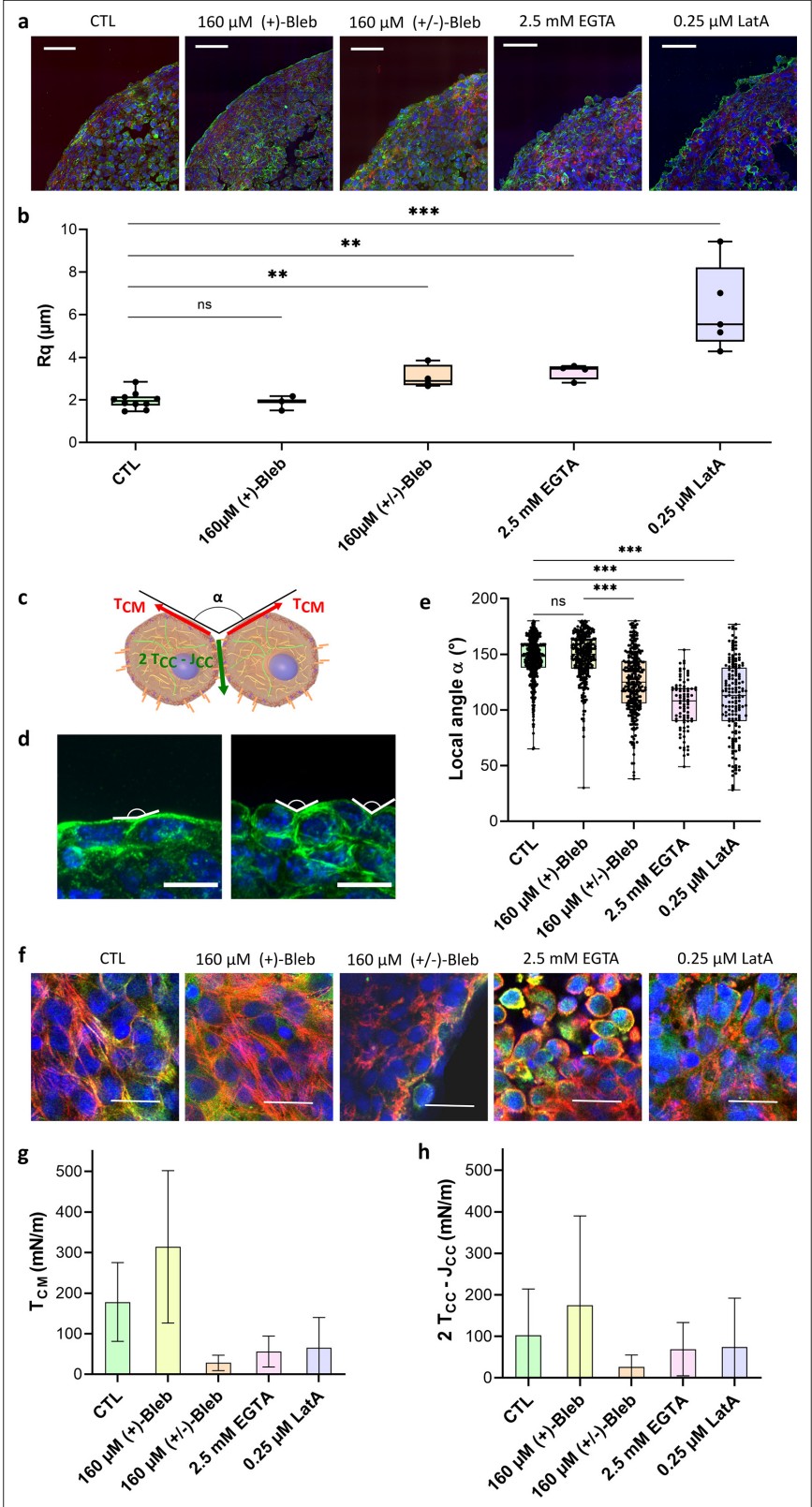

**Figure 3.** Geometrical analysis of cells at the aggregate surface. (**a**) Immunofluorescence images of cryosections of multicellular aggregates obtained by magnetic moulding with different conditions. Control cells are compared to aggregates produced with 160 µM (+)-blebbistatin (inactive enantiomer), 160 µM (±)-blebbistatin (mixture of active and inactive enantiomers), 2.5 mM EGTA or 0.25 µM latrunculin A. DAPI is shown in blue, pan-cadherin in green,

*Figure 3 continued on next page*

*Figure 3 continued*

and F-actin in red. Scale bar: 50 μm. (**b**) Profile surface roughness parameter $R_q$ (root-mean-squared) in each condition for at least $N = 3$ spheroids. (**c**) Schematic of two neighbouring cells with a local contact angle $\alpha$ and the respective tension at the cell–medium interface $T_{CM}$ and effective tension at the cell–cell contact $2T_{CC} - J_{CC}$. (**d**) Examples of immunofluorescence images of spheroids from which the local angles were measured. Nuclei are shown in blue and pan-cadherin in green. Scale bar: 10 μm. (**e**) Contact angle between cell surfaces measured in each condition for at least $N = 3$ spheroids either on cryosections or 3D aggregates (*Figure 3—figure supplement 1*). (**f**) Examples of immunofluorescence images of the upper cells on spheroids obtained with different conditions. Nuclei are shown in blue, actin in red, and phospho-myosin in green. Colocalization of phospho-myosin and actin is in yellow. Scale bar: 25 μm. (**g, h**) Deduced values of the cell tension at the cell–medium interface (**g**) and of the effective adhesive tension at the cell–cell contact (**h**) in each condition.

The online version of this article includes the following source data and figure supplement(s) for figure 3:

**Source data 1.** Source data of the roughness of imaged multicellular aggregates in various conditions presented in *Figure 3a*.

**Source data 2.** Source data of the local contact angles and the tension at the cell–medium and the cell–cell interfaces measured on multicellular aggregates in various conditions reported in *Figure 3e, g and h*.

**Figure supplement 1.** Comparison between local angles measured on spheroid cryosections or 3D images for C2C12 WT spheroids.

on the one hand, and macroscopic surface tension, on the other. Latrunculin A and EGTA-treated aggregates show rounded cells at the interface while control cells flatten on the surface without extending over multiple cells. (±)-Blebbistatin-treated aggregates have an intermediate behaviour. To quantify these observations, we extracted both roughness of the profile and mean contact angle $\alpha$ between cells at the interface (*Figure 3*). They show similar variations. For control aggregates and (+)-blebbistatin aggregates, cells at the surface spread out, having a flat angle at the CC contacts and a small roughness length. Drug drastic effects can be noticed on EGTA and latrunculin A-treated aggregates: roughness increases (by 3 for the latrunculin A treated cells and by 2 for EGTA treated cells) while contact angles deviate from flat angle to get close to 100°. Overall, the surface tension variations are correlated with a change in morphology of the cells at the interface. As already noticed (*Manning et al., 2010*), high tissue surface tension usually appears as the hallmark of flattened cells and low roughness. However, the case of (±)-blebbistatin inhibitor shows that multiple parameters have to be considered as the low surface tension obtained with the myosin inhibitor does not lead to rounded cells at the interface. Tissue surface tension arises from a balance between cortical tension at the CM and adhesion and cortical tension at the CC interface (*Equation 1*; *Manning et al., 2010*; *Stirbat et al., 2013*). Tissue surface tension quantifies this interplay and not only the tension at the CM interface while being predominant. None of the considered aggregates shows elongated surface cells extending over multiple inner cells, meaning that the effective CC interfacial energy $2T_{CC} - J_{CC}$ is smaller than the double cortical tension at the CM interface. In this configuration, differential interfacial tension hypothesis and more sophisticated models provide similar results (*Manning et al., 2010*), and the local mechanical equilibrium at the three phases cell/cell/medium contact line gives the relation (*Stirbat et al., 2013*)

$$2T_{CM} \cos\left(\frac{\alpha}{2}\right) = 2T_{CC} - J_{CC} \tag{2}$$

Both effective CC tension $2T_{CC} - J_{CC}$ and cortical tension at the CM interface $T_{CM}$ can then be deduced from surface tension and contact angle measurements:

$$T_{CM} = \frac{\gamma}{1 - \cos\frac{\alpha}{2}} \quad ; \quad 2T_{CC} - J_{CC} = 2\gamma \frac{\cos\frac{\alpha}{2}}{1 - \cos\frac{\alpha}{2}}$$

First, we checked whether the tension at the CM interface is predominant in the value of surface tension. As noticed on the cryosections, EGTA and latrunculin A impact predominantly the tension at the CM interface with a reduction by a factor of around *Figure 3g*. By reducing CC adhesion, EGTA impairs the formation of contractile actin network at the frontier of the aggregate, and actin is mostly cortical (*Figure 3f*). By depolymerizing actin, latrunculin A reduces global cortical tension as well as the CC adhesion by weakening cadherin anchorage through actin. Feedback between adhesion

molecules and the cytoskeleton is indeed abundant and explains the lower variations of effective CC tension upon latrunculin A and EGTA addition due to a possible compensation of the tension decrease by a decrease in adhesion ($J_{CC}$). Besides, blebbistatin has a significant effect on both effective adhesion and tension at the CM as it impacts neither intercellular adhesions nor the cytoskeleton structure but the ability of the cell cortex to contract. By decreasing tension at the cortex, this inhibitor reduces both $T_{CM}$ and $T_{CC}$, thus affecting tension and effective CC adhesion in a more drastic way than latrunculin or EGTA. Hence, the magnetic tensiometer appears as a highly sensitive tool to look at tissue model mechanics and its relation with modifications at the cellular level.

## Role of intermediate filaments in macroscopic mechanical properties of muscular tissue models

Desmin is an intermediate filament specific to muscle cells where it plays an essential role in maintaining mechanical integrity and elasticity (*Even et al., 2017*) at the single-cell level. It stands as a marker for muscular cell differentiation, but its role in tissue surface tension maintenance and elasticity has not been explored. Desmin mutations are involved in human diseases such as certain skeletal and cardiac myopathies (*Goldfarb et al., 2004*), characterized histologically by intracellular protein aggregates containing desmin. We focus on C2C12 myoblasts expressing desmin with the missense mutation D399Y (*Segard et al., 2013*). We use three cell lines: A21V cells, which are stably transfected with an empty vector and are control cells; desWT-Cl29 cells, which stably express exogenous wild-type (WT) desmin with a ratio of around 1:1 compared to endogenous desmin; and desD399Y-Cl26 cells, which stably express exogenous mutated desmin with a ratio of around 1:1 compared to endogenous desmin (*Delort et al., 2019*). Desmin overexpression and modification do not affect desmin organization of adherent cells in the absence of induced protein aggregation (*Figure 4—figure supplement 1*). Surface tension and Young's modulus are not or hardly impacted by the overexpression of desmin (wild-type or mutated) (*Figure 4f*). Comparing A21V cells to desWT-Cl29, one can notice a slight decrease in elasticity (*Figure 4f*) but it is not measured for desD399Y-Cl26 cells. Besides, as the local contact angle and roughness are similar in all three cell types (*Figure 5a and b*), the deduced CM and effective CC tensions also have values which are not significantly different (*Figure 5e and f*). The three actin networks and the phospho-myosin networks (*Figure 5—figure supplement 1*) are also similar.

In this cellular model, desmin aggregation is induced by heat shock in 2D culture (*Figure 4a–d*, *Figure 4—figure supplement 1*) but also in multicellular spheroids (*Figure 4e*, *Figure 4—figure supplement 2*). In vivo, heat shock is not a common stress, but in skeletal muscle, it models the increase in heat experienced during exercise and fever in case of infection. In desmin mutated cells (desD399Y-Cl26), heat shock duration monitors the percentage of cells presenting desmin aggregates with up to 30% of cells for 2 hr stress compared to a 2% ratio in the case of wild-type desmin expression (desWT-Cl29) (*Figure 4c and d*). As desmin aggregation is induced by heat shock in this model, the effect of protein aggregation and of the physical inducer has to be carefully deciphered. Measurements on A21V control cells allow the effect of a heat shock on surface tension and Young's modulus to be verified. Heat shock decreases both Young's modulus and surface tension of A21V control aggregates (*Figure 4g*, *Figure 4—figure supplement 3*). Conversely, heat shock does not impact roughness and local contact angles between cells at the periphery of the A21V multicellular aggregates (*Figure 5a and b*). The decrease in surface tension therefore translates at the cell tension level and both CM and CC tensions decrease (*Figure 5e and f*). This result is corroborated by the observation of actin network at the periphery of the multicellular aggregates: heat shock does not drastically modify actin distribution but slightly decreases the number of stress fibres (*Figure 5c*). The effect of heat shock on the properties of aggregates of cells overexpressing WT desmin (desWT-Cl29) is similar to that observed for A21V control cells (*Figure 4g*, *Figure 4—figure supplement 3*), but different for aggregates of cells overexpressing mutated desmin (desD399Y-Cl26) (*Figure 4g*, *Figure 4—figure supplement 3*). Focusing on desmin organization, we compare cells expressing mutated desmin (desD399Y-Cl26) with cells expressing wild-type desmin (desWT-Cl29). An increase in surface tension and elasticity dependent on the heat shock duration is observed in cells overexpressing mutated desmin (*Figure 4f and g*): the surface tension of spheroids of desD399Y-Cl26 cells is 1.9-fold higher than the one of desWT-Cl29 cells and their elasticity is 2-fold higher for 30 min heat shock. The effect is even greater for a 2 hr heat shock with a two times higher surface tension of multicellular aggregates and a three times higher Young's modulus for cells expressing mutated desmin compared to

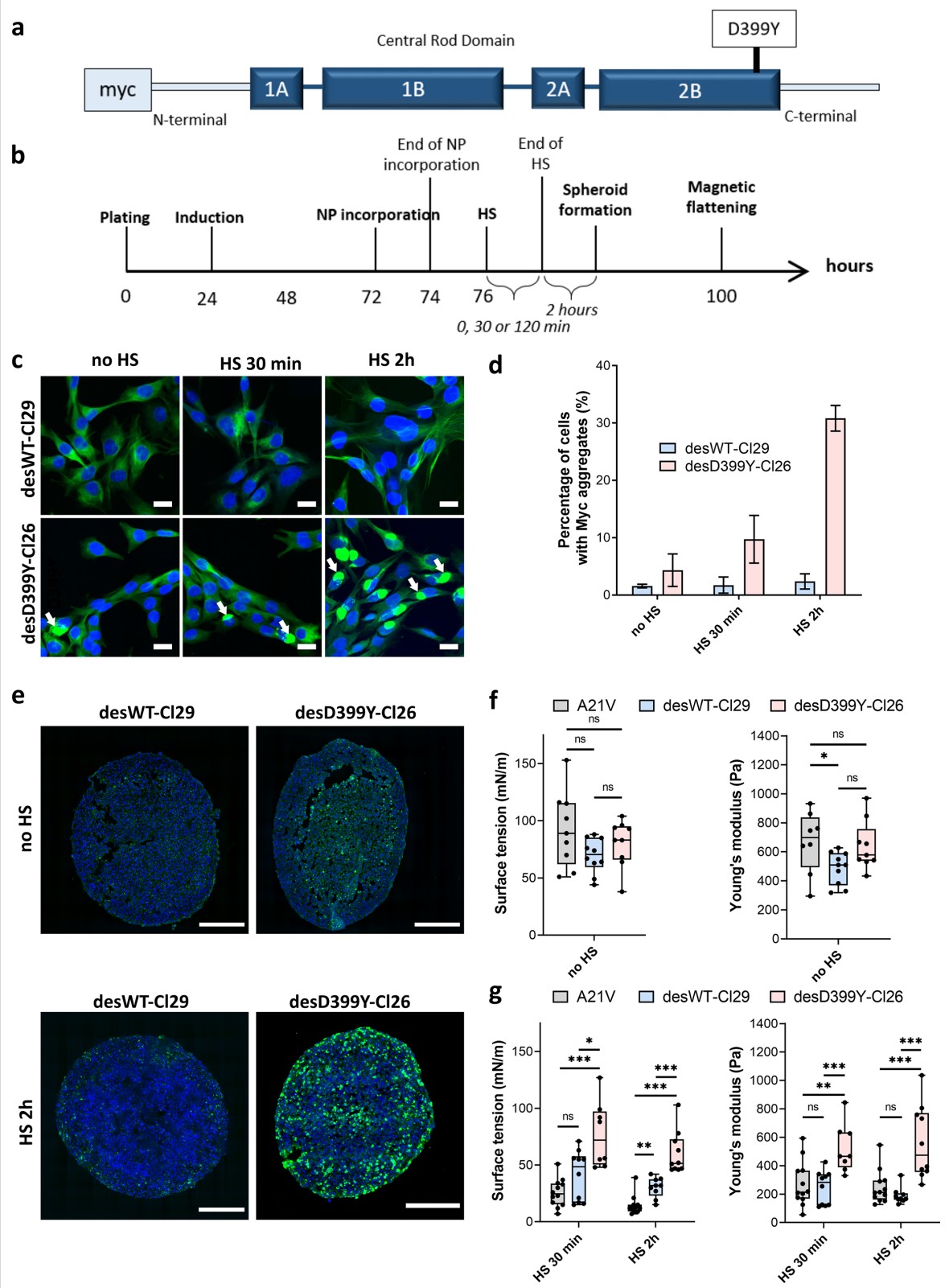

**Figure 4.** Effect of heat shock (HS) and protein aggregation of desmin on surface tension and Young's modulus of C2C12 spheroids. (**a**) Representation of desmin with the missense mutation D399Y located in Rod Domain. The expressed exogenous mutated desmin is Myc-tagged at the N-terminus (adapted from Figure 1 from *Segard et al., 2013*). (**b**) Experimental procedure. Desmin expression was induced 24 hr after cell plating with doxycycline for 48 hr. Magnetically labelled cells (see 'Methods') experienced an HS for 0, 30, or 120 min before spheroids were moulded. Spheroid

*Figure 4 continued on next page*

*Figure 4 continued*

surface tension and Young's modulus were measured 3 days after expression induction. (**c**) Immunofluorescence images of desWT-Cl29 (cells stably expressing exogenous desmin WT) and desD399Y-Cl26 cells (cells stably expressing exogenous mutated desmin) in 2D after 0, 30, or 120 min HS. DAPI is shown in blue and the Myc-tag in green. Scale bar: 20 μm. (**d**) Percentage of cells with desmin protein aggregates for each condition. Desmin protein aggregation increases for desD399Y-Cl26 cells with the duration of the HS while it remains stable around 2% for desWT-Cl29 cells. Mean values represented with respective standard deviations and at least three independent experiments for each condition. (**e**) Immunofluorescence images of multicellular aggregate cryosections of desWT-Cl29 or desD399Y-Cl26 cells with or without HS. DAPI is visible in blue and the Myc-tag in green. The results are reminiscent of the ones in 2D. DesD399Y-Cl26 spheroids exhibit sparse aggregation without HS, enhanced by 2 hr HS. Scale bar: 200 μm. See *Figure 4—figure supplement 2*. (**f**) Surface tension and Young's modulus of A21V cells (control cells stably transfected with empty vector) spheroids compared with desWT-Cl29 and desD399Y-Cl26 cells to test for the influence of desmin overexpression. (**g**) Surface tension and Young's modulus of A21V cells, desWT-Cl29, and desD399Y-Cl26 spheroids with an HS of 0 (no HS), 30 or 120 min. (**f, g**) At least three independent experiments for each condition and $N = 8$ spheroids. Floating bars represent min to max variations, and the midline indicates the median.

The online version of this article includes the following source data and figure supplement(s) for figure 4:

**Source data 1.** Source data for the surface tension and Young's modulus measurements of aggregates shown in *Figure 4d, f and g*.

**Figure supplement 1.** Desmin organization Immunofluorescence images of desmin on adherent cells of control A21V, desWT-Cl29, and desD399Y-Cl26 cell lines for no heat shock (HS) or 2 hr HS.

**Figure supplement 2.** Immunofluorescence images of C2C12 spheroid cryosections of desWT-Cl29 (**a, c**) and desD399Y-Cl26 cells (**b, d**) for no heat shock (HS) (**a, b**) or 2 hr HS (**c, d**).

**Figure supplement 3.** Surface tension and Young's modulus of A21V spheroids (**a**), of desWTCl29 spheroids (**b**) and desD399Y-Cl26 (**c**) with no heat shock (HS), HS 30 min or HS 2 hr.

cells expressing the wild-type desmin. The effects on mechanical properties are therefore correlated with the percentage of cells containing desmin aggregates. Local cell disorganization modifies surface tension probably through individual cell tension modifications.

Looking at the local geometry at the surface of multicellular aggregates, cells overexpressing desmin also show a predominance of tension at the CM interface (*Figure 5*). When desmin overexpression is induced, cortical tension at the CM interface slightly increases (by a factor of 2) comparing desWT-Cl29 and desD399Y-Cl26 spheroids with a 2 hr HS while effective adhesion is less impacted. Overexpression of wild-type and mutated desmin does not change phospho-myosin distribution on the overall aggregate (*Figure 5—figure supplement 1*) and desmin aggregates are not colocalized with contractile points (*Figure 5*, *Figure 5—figure supplement 1*). However, mutated desmin aggregation strengthens long-range phospho-myosin filaments at the periphery of the spheroid (*Figure 5d*) which increases acto-myosin network contractility.

## Discussion

Magnetic tensiometry gives a high sensitivity and precision even on a cellular system that does not easily deform as muscle cells. It thus pushes back the limits of macroscopic mechanics sensitivity to molecular modifications. In this study, the values obtained for cellular tensions either at the CM interface or at the intercellular one confirm this specificity as tensions are 2–3 orders of magnitude higher than the ones measured in 3D F9 aggregates (*Stirbat et al., 2013*) or in embryos (*Maître et al., 2012*; *Maître et al., 2015*). This result is not surprising as spheroids of C2C12 cells are organized in a highly contractile structure with a high density of stress fibres at the periphery. This tensile network makes them challenging to characterize, thus requiering sensitive tools. Magnetic tensiometer sensitivity even for these high-surface tension model tissues is in the range of the one obtained with compression plates tensiometer experiments for lower surface tension cell aggregates (*Mgharbel et al., 2009*) and proves to be more accurate for more deformable spheroids (*Mazuel et al., 2015*). The accessible range of measurable surface tension with the magnetic tensiometer is therefore larger.

In addition, the magnetic tensiometer measurements make it possible to provide dose–response curves, thus representing an accurate tool to quantitatively determine inhibitors' action on mechanical properties. Its robustness is related to the spheroid formation technique that allows providing well-controlled aggregates of unprecedented size with radius and content perfectly monitored in less than 24 hr (*Mazuel et al., 2015*). It offers a unique opportunity to explore mechanical properties even in a dose-dependent manner and quantitatively extract the effects of mutations.

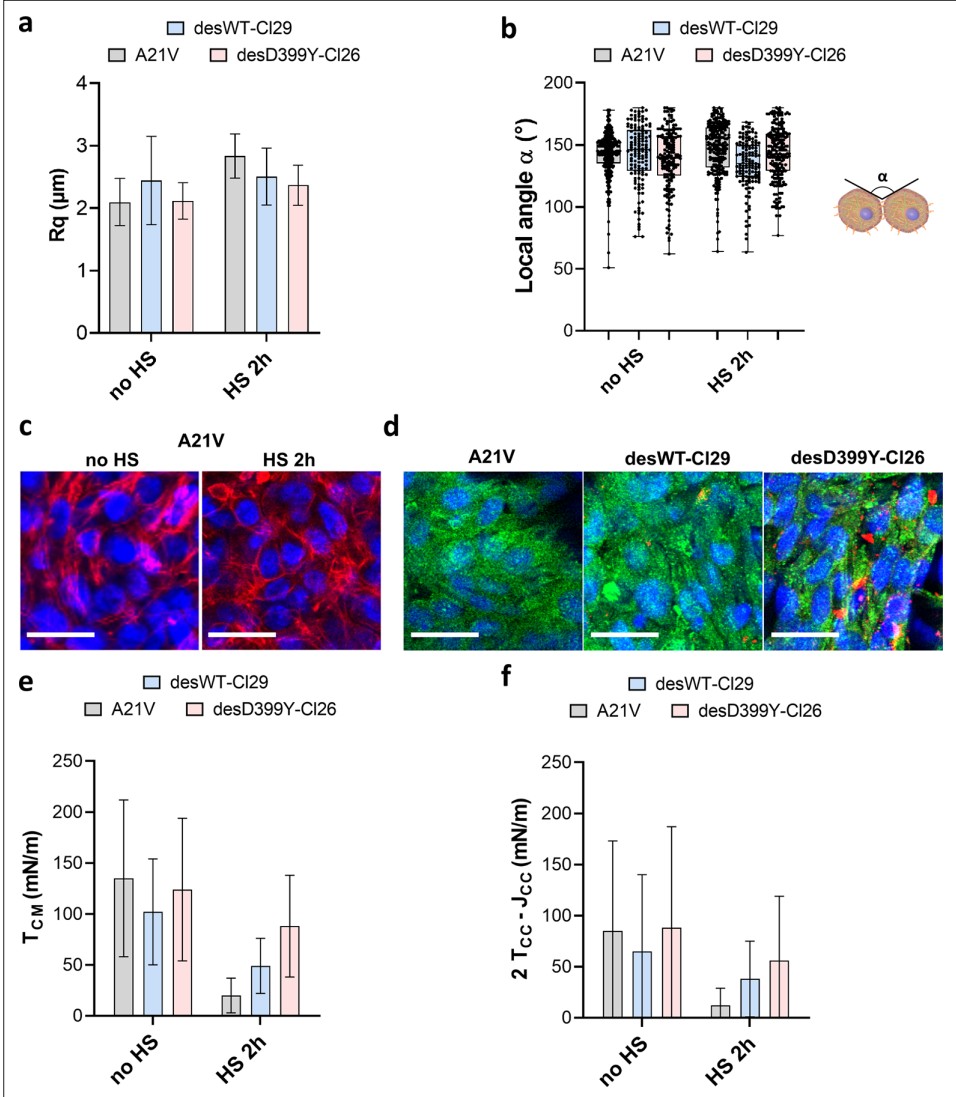

**Figure 5.** Geometrical analysis of cells at the aggregate surface for C2C12 A21V, desWT-Cl29, and desD399Y-Cl26 spheroids. (**a**) Profile surface roughness parameter Rq (root-mean-squared) in each condition for at least $N = 3$ spheroids. (**b**) Local contact angle between cells at the surface measured in each condition for at least $N = 3$ spheroids. (**c**) Representative confocal images of actin network for the A21V cells at the surface of the multicellular aggregate with or without HS. Nuclei are labelled in blue and F-actin in red. Scale bar: 25 µm. (**d**) Representative confocal images of phospho-myosin and desmin aggregates distribution for cells at the surface of multicellular aggregates. A21V cells are compared to cells overexpressing desmin (desWT-Cl29 and desD399Y-Cl26) after 2 hr heat shock. Nuclei are labelled in blue, phospho-myosin in green, and desmin aggregates in red. Scale bar: 25 µm. (**e, f**) Deduced values of the cell tension at the cell–medium interface (**f**) or of the effective tension at the cell–cell contact (**g**) in each condition.

The online version of this article includes the following source data and figure supplement(s) for figure 5:

**Source data 1.** Source data of the roughness of multicellular aggregates for control A21V, desWT-Cl29, and desD399Y-Cl26 cells presented in *Figure 5a*.

**Source data 2.** Source data of the local contact angles measured for control A21V, desWT-Cl29, and desD399Y-Cl26 cells presented in *Figure 5b, e and f*.

**Figure supplement 1.** Quantification of phospho-myosin distribution in the spheroids.

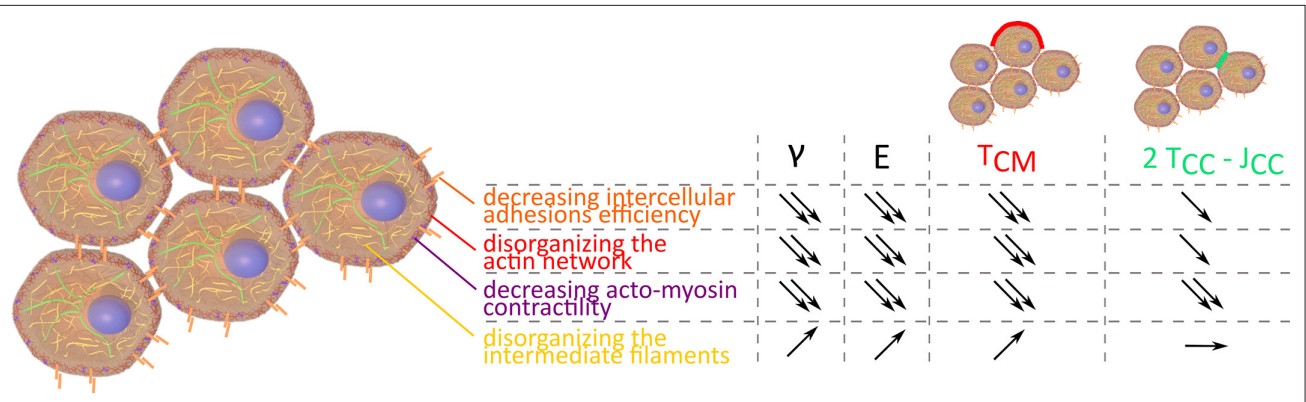

**Figure 6.** Evolution of $\gamma$, E, $T_{CM}$, and $2\,T_{CC} - J_{CC}$ depending on intercellular adhesions, actin network, acto-myosin contractility, and intermediate filaments.

Measuring surface tension and elasticity proves to be a powerful tool to explore tissue mechanical properties. Both are key elements of tissue shape maintenance (*Dahmann et al., 2011*; *Kim et al., 2021*). Looking at 3D unorganized model tissues, we recapitulate the major cellular factors that influence tissue shape and response to perturbations. Unsurprisingly, intercellular adhesions and actin cortex structure or contractility are fundamental as was already demonstrated in vivo (*Käfer et al., 2007*; *Iijima et al., 2020*). Their decrease reduces both surface tension and elasticity (*Figure 6*). In this study, the major contribution of cell contractility in tissue shape maintenance is pointed out regarding the resulting decay obtained for surface tension and stiffness at high concentrations of (+/-)-blebbistatin. Inhibition at the microscopic level correlates with surface tension and elasticity evolution, confirming them as relevant indicators of molecular state and cellular contractility checked by looking at acto-myosin network. The crucial role of acto-myosin contractility is also corroborated by microscopic observations. Long-range actin network at the periphery of the multicellular aggregates is fundamental to macroscopic shape maintenance while the slightest modification of this network drastically changes this property.

Altogether, our results suggest that measuring mechanical properties at tissue scale provides insights into the molecular level (*Gradeci et al., 2021*) and that, conversely, molecular modifications can induce mechanical changes at tissue scale. Complemented by local measurements of the contact angle of cells at the surface of our multicellular aggregates, our approach is able to determine the relative influence of cytoskeleton structure, adhesion molecules, and cortical tension on the surface tension and may distinguish tension from adhesion modifications.

Desmin is the most representative intermediate filament for muscle cells. It has a pivotal role in myofibril architecture in mature muscle but also exerts important function on the adaptation of muscle cells to passive stretch and contractility (*Diermeier et al., 2017*). Desmin organization defects severely impact muscle formation and maintenance, causing myopathies or cardiomyopathies (*Batonnet-Pichon et al., 2017*). In desminopathies, desmin aggregates are characteristic of the pathology; however, these aggregates remain heterogeneously distributed within the cardiac or skeletal muscle tissues. Furthermore, the adult onset of the disease and the wide spectrum of phenotypes point to a sensitivity to the environmental factors. Heat shock taken as an exogenous stress in our cell model manages to mimic heterogenous desmin aggregation in model tissues. However, at the same time, it is documented that the heat stress chosen as a transducer of desmin aggregation in desD399Y cells can affect the biomechanical properties of skeletal muscle cells or myofibres (*Locke and Celotti, 2014*). In our experiments, we reproduce these results as heat shock decreases both surface tension and Young's modulus for cells without desmin aggregation by impacting actin contractility. This mechanism observed in vivo may be seen as a protective effect to avoid muscle damage. Desmin aggregation seems to impair this mechanism and is potentially the signature of a lack of cell adaptation.

Furthermore, in single cells, desmin aggregation has been associated with sarcomere misalignment but also impacts biomechanical properties (*Clemen et al., 2013*). We measure its action on surface tension and elasticity: desmin aggregation leads to an increase in both elasticity and surface tension. Desmin acts on muscle functioning and integrity (*Charrier et al., 2018*; *van Bodegraven and*

*Etienne-Manneville, 2021*), and its disorganization may hinder individual cell contractility (*Franck et al., 2019*) and homeostasis (*Agnetti et al., 2022*). Our results highlight its major role at the tissue level where it impacts cell tension mainly at the CM interface by increasing cell contractility. In zebrafish, it was shown that invalidation of the *snaib* factor induced extrusion of cardiomyocytes with increased contractility and an alteration in the *desmb* gene (orthologue of the *des* gene in humans) (*Gentile et al., 2021*). Overexpression of desmin recapitulates this process and highlights its role in maintaining the myocardial wall. In our model, desmin does not increase but is mutated, and this mutation induces desmin aggregation. However, the same increase in contractility which may lead over time to myofibril fragility could have an impact on sarcomere alignment and force transmission in cardiac or skeletal muscle in myopathies. Intermediate filament organization therefore appears essential in muscle tissue shape maintenance and mechanical properties even at an early stage of differentiation. At the individual scale, mutated desmin cells show modified reorientation dynamics (*Leccia et al., 2013*) and are slightly (but not significantly) stiffer (*Even et al., 2017*), a behaviour reminiscent of the elasticity increase noticed on mutated desmin filaments D399Y (*Kreplak and Bär, 2009*). However, 2D studies fail to reproduce the severity of the effects on tissue mechanics. Conversely, stiffening is translated at the multicellular scale on the Young's modulus (*Figure 4g*) but the effect is enhanced. While a 1.4-fold increase was noticed in single-cell elasticity, macroscopic Young's modulus of multicellular aggregates significantly increased by 3-fold when comparing mutated to wild-type desmin. Moreover, while desmin overexpression does not affect surface tension (*Figure 4f*), desmin protein aggregation raises its value (*Figure 4g*). By giving more rigid network nodes, protein aggregates actually strengthen cell cortex (*Even et al., 2017*) and may reinforce effective cortex tension, thus increasing tissue surface tension. Compared to wild-type desmin, they enhance macroscopic long-range contractile network efficiency.

We demonstrate that desmin aggregation impacts both elasticity and surface tension of myoblast spheroids, pointing out the importance of desmin organization on macroscopic tissue mechanics even at an early stage of differentiation. Surface tension and elasticity are thus sensitive reporters of the individual cellular state but also of the multicellular organization. Two reasons may be involved. First, spheroids are made of about 150,000 cells; measuring macroscopic properties allows the properties of individual cells to be averaged over a large population. Secondly, collective effects appear as the change in cell organization modulates both CM and CC tensions, both of which are involved in surface tension (*Stirbat et al., 2013*). Desmin maintains the shape of multicellular aggregate, its role on individual cell is translated, and its action is enhanced at the tissue level.

## Conclusion

Macroscopic properties of multicellular spheroids such as surface tension and elasticity appear as highly sensitive markers for cell cortex and CC adhesion modifications. Measuring them by an approach based on magnetic cell labelling and multicellular aggregate tensiometry allows exploring dose–response evolution correlated with microscopic inhibition potency. The precision provided by the magnetic tensiometer opens a new field of investigation to test the impact of potential drugs or genetic modifications on the mechanics of early-stage tissue models that can be extended to little deformable cell types. We first identified the usual suspects of cellular mechanics (actin network, actomyosin contractility, and CC adhesion) as being fundamental in tissue surface tension and elasticity, but our approach was also used on a cellular model of desmin-related myopathies. It demonstrated the crucial role of intermediate filaments in tissue shape maintenance and Young's modulus. Desmin disorganization induces macroscopic changes in early-stage tissue models, detected only at the individual cellular scale. Like actin or cadherins, desmin acts as an important integrator of tissue shape maintenance. The fully integrated system of magnetic moulding and magnetic tensiometry can be envisioned as a powerful tool for the study of fundamental biological processes and the detection of mechanical effects, leading to a better understanding of skeletal muscle dystrophies.

## Methods
### C2C12 cell culture
C2C12 WT (CRL-1772; RRID:CVCL_0188) were obtained from ATCC and cultured in Dulbecco's modified Eagle's medium (DMEM, Gibco), supplemented with 1% Penicillin-Streptomycin (P/S, Gibco) and

10% fetal bovine serum (FBS, Gibco). Their identity has been authenticated by COI assay. C2C12 A21V (stably transfected with empty vector), desWT-Cl29 (stably expressing exogenous human WT desmin), and desD399Y-Cl26 (stably expressing exogenous human D399Y desmin) were cultured in 1% P/S, 20% FBS, 1 mg/mL geneticin (10131027, Gibco) and 2 µg/mL puromycin (P7255, Sigma-Aldrich) as described in *Segard et al., 2013*. All cell lines were tested negative for mycoplasma contamination.

## Magnetic labelling

Iron oxide superparamagnetic nanoparticles (8 nm diameter) (provided by PHENIX Laboratory, UMR 8234, Paris) were obtained by alkaline coprecipitation followed by oxidation into maghemite according to *Massart, 1981* procedure. Finally, the aqueous solution was stabilized electrostatically by the adsorption of citrate anions on the nanoparticles surface. Cells were magnetically labelled with an incubation of 2 hr with a solution of iron oxide nanoparticles at $[Fe] = 4$ mM and supplemented with 5 mM citrate in RPMI medium (Gibco). The viability and proliferation of cells after magnetic labelling were assessed by Alamar Blue showing no impact of the magnetic labelling right after the labelling or after 1 day (*Figure 1—figure supplement 1*).

## Magnetic moulding

Labelled C2C12 cells were incubated for at least 2 hr in complete medium. Agarose moulds were previously prepared by heating a solution of 2% agarose (A0576, Sigma-Aldrich) in phosphate-buffered saline (PBS) up to boiling. The resulting solution was then poured into a 60 mm culture Petri dish containing 1.6 mm steel beads (BI 00151, CIMAP) held in place by magnets below the Petri dish. Once the agarose solidified, the beads were removed carefully and the agarose wells were coated with a non-adhesive coating solution (07010, Stemcell Technologies) for 30 min at room temperature (RT). Cells were then detached with 0.05% Trypsin-EDTA and seeded in the wells using magnet attraction (about $1.5 \times 10^5$ cells per well). Spheroids were then incubated overnight at 37°C, 5% $CO_2$ in complete medium in normal conditions. At this point, the culture medium may be modified to add inhibitors (*Figure 1*). Spheroids were then extracted from the wells by gently pipetting the surrounding liquid. Spheroids of 530 ± 60 µm radius (N = 167) were obtained. By measuring the magnetic moment of the aggregate with vibrating-sample magnetometer (VSM) measurements (PPMS – Quantum Design), the magnetic moment per unit volume of the aggregate $M_v$ can be determined and the value of surface tension deduced. $M_v$ measurements ranged from 150 to 500 A.m$^{-1}$ and were determined for each experiment. The force per unit of volume exerted on the spheroids ranged from $2.5 \times 10^4$ to $8.5 \times 10^4$ N.m$^{-3}$.

## Magnetic force tensiometer

The magnetic force tensiometer is composed of a temperature-regulated tank at 37°C sealed at the bottom and the sides by glass slides, a 6 × 6 mm cylindrical neodymium permanent magnet (S-06-06N, Supermagnete, $B = 530$ mT, $grad(B) = 170$ T/m, averaged value between 250 µm and 1.75 mm from the surface of the magnet), a lifting stage to approach the magnet and a camera (QICAM FAST1394, QImaging) equipped with a ×1.5 zoom lens (MVL6X12Z, Thorlabs). The temperature-regulated tank is filled with a transparent culture medium (DMEM, high glucose, HEPES, no phenol red, Gibco). The bottom slide is treated with a non-adhesive solution (07010, Stemcell Technologies) to guarantee non-wetting conditions for the multicellular spheroids. Horizontality of the bottom slide has to be carefully checked. The aggregate side profile is registered with the camera. The magnet is approached at 150 µm from the bottom of the aggregate. The macroscopic mechanical properties of the spheroid are obtained by the spheroid profile at equilibrium (10 min under magnetic flattening to ensure that equilibrium is reached). Surface tension is measured by fitting the spheroid profile at equilibrium with Laplace profile while Young's modulus is determined with respect to the radius of the contact zone $L$ and using Hertz theory as described in *Mazuel et al., 2015*. The Young's modulus $E$ equals $E = \frac{(1-\sigma^2)\pi M_v \, grad(B) R^4}{L^3}$, where $\sigma$ stands for the Poisson ratio ($\sigma = 1/2$), $M_V$ represents the magnetic moment per unit volume, B the magnetic field and $R$ and $L$ are the initial radius and radius of the contact zone, respectively. Briefly for the surface tension measurements, theoretical profiles were obtained by integrating numerically the Laplace law for capillarity and minimizing the quadratic error on the height $h$, width $w$, and volume $V$ of the spheroid (*Figure 1—figure supplement 3*) to extract the capillary constant $c = \frac{M_v \, grad(B)}{\gamma}$ (*Kalantarian et al., 2009*). Depending on the conditions,

flattening occurs either with a two times regime or exhibits a single time decay. Spheroids have two ways of deformations to reach their capillary-driven equilibrium shape: first, a rapid elastic deformation, then a more viscous fluid behaviour. This competition is driven by the size of the aggregate, meaning that above a critical radius $R_C$ (*Mazuel et al., 2015*), elastic deformation is complete and may be followed by viscous-like behaviour to reach capillary-driven limit. Here, all the measurements were done with a spheroid radius in the range of this critical radius so that Young's modulus can be extracted. This modulus is extracted at the end of the elastic deformation while the surface tension is deduced from equilibrium shape. Tensio𝕏 is a dedicated MATLAB-generated application freely available to extract both surface tension and Young's modulus from initial and flattened profiles. Images of the initial and final state are first downloaded. The user has to define the initial radius, height, and width by pointing left/right and top/bottom frontiers of the spheroid on the two images. The radius of the contact zone is also extracted. The scale factor, magnetic force per unit volume, and an estimated surface tension value are entered. The obtained fit is then superimposed on the flattened spheroid image to check for its relevance and the deduced Young's modulus and surface tension are registered in a separate data file.

## Measurements of mechanical properties of C2C12 spheroids

### In drug conditions

For inhibitor conditions, the drug was added to the medium when the cells were seeded in the wells. C2C12 spheroids were then incubated overnight at 37°C, 5% $CO_2$ in DMEM 10% FBS, 1% P/S supplemented with the chosen concentration of EGTA (03777, Sigma-Aldrich), latrunculin A (L5163, Sigma-Aldrich), (+)-blebbistatin (203392, Sigma-Aldrich) or (±)-blebbistatin (203390, Sigma-Aldrich). For (±)-blebbistatin, inhibition curves were fitted with the function $Y(X) = Bottom + \frac{(Top - Bottom)}{1 + \frac{IC_{50}}{X}^{HillSlope}}$ and with no constraints applied to the four variable parameters.

### C2C12 expressing mutated desmin

At day 0, cells were plated at $3000 \, \text{cells/cm}^2$ ($2.25 \times 10^5$ cells seeded in a T-75 flask). After 24 hr of incubation at 37°C, 5% $CO_2$, expression of exogen desmin was induced by supplementing the culture medium with 10 µg/mL doxycycline (D9891, Sigma-Aldrich) for 2 days. The medium was replaced with fresh medium every 24 hr. Iron oxide nanoparticles were incorporated into the cells on day 3. After 2 hr of recovery, a heat shock (HS) with a water bath at 42°C for 0, 30, or 120 min was applied to the cells. 2 hr later, cells were detached and used to form spheroids by magnetic moulding or seeded in 24-well plates for desmin aggregation evaluation. After an overnight incubation at 37°C, 5% $CO_2$, spheroids were magnetically flattened to measure their mechanical macroscopic properties, then fixed for 1 hr at room temperature in 4% paraformaldehyde (PFA, J61899, Alfa Aesar). The corresponding cells grown in 2D in the 24-well plates were fixed for 15–20 min at RT in 4% PFA to assess the percentage of cells with Myc aggregation for each condition.

### Cryosections and immunofluorescence

Spheroids were fixed in 4% PFA for 1 hr at RT and stored in PBS at 4°C. For cryosections, spheroids were embedded in OCT (Optimal Cutting Compound, 361603E, VWR) for 1 hr at RT, they were then frozen in iso-pentane (24872.260, VWR), cooled down in liquid nitrogen, and then stored at -20°C. 6–10 µm cryosections were obtained (Leica CM1520). For immunofluorescence labelling, cryosections or fixed cells were permeabilized 15–20 min in 0.1% Triton X-100 at RT while whole aggregates were permeabilized 1 day in 1% Triton X-100 at 4°C. Non-specific interactions were prevented by an incubation with 5% BSA (#05479, Sigma-Aldrich) for 1 hr at RT (increased to 2 days at 4°C for whole aggregates). Pan-cadherin (rabbit anti-pan cadherin [dilution 1:100 in PBS 0.5% BSA, C3678, Sigma] for 2 hr at RT), phospho-myosin (rabbit phospho-myosin light chain 2 [Thr18/Ser19] antibody [dilution 1:50 in PBS 0.5% BSA, #3674, Cell Signaling] for 2 hr at RT), and desmin (dilution 1:100, in PBS 0.5% BSA, [D8281 or SAB4200707, Sigma] for 2 hr at RT) were labelled. The two first primary antibodies were coupled with an Alexa Fluor 488 goat anti-rabbit secondary antibody (dilution 1:500 in PBS 0.5% BSA, #4412, Cell Signaling Technology) for 2 hr at RT while the last one was coupled to an Alexa Fluor 555 anti-mouse secondary antibody. Myc was labelled using mouse c-Myc (9E10) Alexa Fluor 647 (dilution 1:100 in PBS 0.5% BSA, sc-40AF647, Santa Cruz Biotechnologies) for 2 hr at RT. F-actin

was labelled using SiR-actin or SPY555-actin (dilution 1:1000 in PBS, Spirochrome) for 1 hr 30 min at RT, while nuclei were labelled with DAPI (dilution 1:1000 in PBS, D9564, Sigma-Aldrich) or Hoechst 33342 (dilution 1:1000 in PBS, H3570, Invitrogen) for 15–20 min at RT. All the samples were mounted with Fluoromount (F4680, Sigma-Aldrich) and stored at 4°C after gelation of the mounting medium at RT. Labelling of the whole aggregates was done in the same conditions, but the incubation times were extended to 2 days at 4°C. Cryosections were imaged either on a Nikon microscope A1r25HD with a ×100 oil objective or on an Axio observer Zeiss microscope equipped with a CSU-X1 Spinning disk with a ×63 oil objective. Whole aggregates were imaged on a Zeiss 780 confocal microscope equipped with a ×20 water immersion objective.

### Roughness measurements

The contour profile of the spheroids was extracted manually with Fiji (ImageJ) from immunofluorescence images of spheroid cryosections (pan-cadherin, F-actin, or phospho-myosin). The extracted experimental profile was then fitted by a circular arc and the roughness parameter $R_q$ was measured by computing for each experimental point the distance $z$ to the circular arc and then calculating $R_q = \frac{1}{N}\sqrt{\sum_i z_i^2}$ with $N$ being the number of points of the experimental contour. The smaller $R_q$ is, the less rough the surface of the spheroid is, and the closer it is to a circular arc.

### Local contact angle measurements

Local contact angles were measured with respect to pan-cadherin immunofluorescence images of spheroid cryosections or spheroids imaged in 3D. Contact angles between cells at the spheroid surface were measured using Fiji (ImageJ). Pan-cadherin labelling was used to confirm the adhesion between two neighbouring cells for each measurement. Measurements were performed on at least three spheroids for each condition, and angles were measured all over the surface of the spheroid. No significant differences were noticed regarding the imaging source (cryosections or spheroids) as evidenced in *Figure 3—figure supplement 1*.

## Statistical analysis

All statistical tests were performed with a two-sided Mann–Whitney U test (Wilcoxon test) using MATLAB. p-Values are used to indicate the statistical significance of the results (*, **, and *** correspond to p<0.05, p<0.01, and p<0.001, respectively).

## Acknowledgements

The authors thank Alexandre Fromain for his help on VSM measurements and Ali Abou Hassan for providing us magnetic nanoparticles. This work was supported by the Program Emergence(s) de la Ville de Paris (Grant MAGIC Project). The study was partially supported by the Labex Who Am I?, Labex ANR-11-LABX-0071, the Université de Paris, Idex ANR-18-IDEX-0001 funded by the French Government through its Investments for the Future program, the AFM (French Association for Myopathies) AFM-22956, and the French Defense Procurement Agency (DGA-AID) France. We acknowledge the ImagoSeine core facility of the Institute Jacques Monod (member of the France BioImaging, ANR-10-INBS-04) and France Lam at the Cellular Imaging facility of IBPS for her advice on deep penetration tissue imaging. We thank the staff of the MPBT (physical properties – low temperature) platform of Sorbonne Université for their support.

## Additional information

### Funding

| Funder | Grant reference number | Author |
|---|---|---|
| Emergence(s) Ville de Paris | MAGIC | Myriam Reffay |
| Association Française contre les Myopathies | AFM-22956 | Sabrina Batonnet-Pichon |

| Funder | Grant reference number | Author |
|---|---|---|
| Direction Générale de l'Armement | | Myriam Reffay |
| Labex Who Am I? | Labex ANR-11-LABX-0071 | Myriam Reffay |

The funders had no role in study design, data collection and interpretation, or the decision to submit the work for publication.

### Author contributions

Irène Nagle, Resources, Software, Formal analysis, Investigation, Methodology, Writing - original draft; Florence Delort, Investigation, Methodology, Writing – review and editing; Sylvie Hénon, Conceptualization, Methodology, Writing – review and editing; Claire Wilhelm, Supervision, Writing – review and editing; Sabrina Batonnet-Pichon, Conceptualization, Supervision, Methodology, Project administration, Writing – review and editing; Myriam Reffay, Conceptualization, Resources, Data curation, Software, Formal analysis, Supervision, Funding acquisition, Investigation, Methodology, Writing - original draft, Project administration

### Author ORCIDs

Myriam Reffay http://orcid.org/0000-0002-3695-2789

### Decision letter and Author response

Decision letter https://doi.org/10.7554/eLife.76409.sa1
Author response https://doi.org/10.7554/eLife.76409.sa2

## Additional files

### Supplementary files

• Transparent reporting form

### Data availability

Data supporting the findings of this study are available within the article and its Supplementary information files. Computing resources should be found on Github (https://github.com/mreffay/INagle-MReffay, copy archived at swh:1:rev:fa38dea19f2693c0306e8e615e29b75c5cfe2fd0).

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
