## [Editor Report]

This important work studied the determinants of key physical properties of multicellular assemblies using magnetic flattening of spheroids. The key and convincing result is that intermediate filaments could also be implicated in the setting of the elastic properties of these assemblies, shedding light on this central cellular component and how their modifications could be important to the understanding of some pathologies.

---

## [Decision Letter]

**Decision letter after peer review:**

Thank you for submitting your article "Multiparameters dependance of tissue shape maintenance in myoblasts multicellular aggregates: the role of intermediate filaments" for consideration by *eLife*. Your article has been reviewed by 3 peer reviewers, and the evaluation has been overseen by a Reviewing Editor and Anna Akhmanova as the Senior Editor. The following individual involved in the review of your submission has agreed to reveal their identity: Morgan Delarue (Reviewer #2).

Essential revisions:

1) The determination of γ and the Young's modulus relies on analysis of the aggregate shape. What are the assumptions on the structure of the aggregate and are these fulfilled? It looks like aggregate mechanics are dominated by the first few rows of cells at the surface. What assumptions are made about the aggregate interior? In many images like Figures3a, 4e, S2b, S6 or S7, it looks like the cells do not uniformly fill the aggregate interior and that there are clear gaps between cells. Is this an artefact of the fixation and cryosectioning? Or do the cells in the interior not form a cohesive material? If the cells do not form a cohesive interior, this may explain why the surface tension and the apparent elasticity of the aggregates are so tightly coupled (Figure S5). It would be useful to image some aggregates with live confocal microscopy or light sheet microscopy.

2) The link between single cell properties and aggregate mechanics could be strengthened. The mechanical measurements are done on the aggregates and the single cell mechanical changes are mostly inferred from γ using the angle of contact. So there no independent determinations of Tcc or Tcm. The changes in cytoskeletal organisation remain very qualitative and could be exploited by systematically quantifying the image data to show changes in F-actin, cadherin, and phosphomyosin. For example, the authors claim that desmin aggregation increases surface tension and cortical tension. They could determine if desmin aggregation leads to an increase in phosphomyosin in those locations. The methods indicate that phosphomyosin staining has been done but this appears nowhere in the figures – yet this is likely an important piece of data. On page 5, the authors suggest that the low change in effective adhesion in response to latA and EGTA addition may be due to an increase in adhesion. This could be verified and quantified from the immunostainings.

3) The main conclusions are largely confirmatory except for the role of desmin. The effects of decreased myosin contractility, depolymerisation of F-actin, and decreasing intercellular adhesion are largely expected. We find the data about Desmin's role confusing. In particular, the authors do not compare the mechanical properties of aggregates pre vs post HS. Yet, these comparisons seem to go against their main conclusion that desmin aggregation increases surface tension. In Figure 4f, we can see that prior to heat shock, γ is about 75mN/m in the two desmin lines and E is about 500-600Pa. After heat shock, the surface tension of the WT desmin decreases to 40-50 mN/m but the D399Y (that shows increases aggregation) does not change compared to no HS. Similarly, after HS, E for the WT desmin decreases to 300 Pa while for the D399Y it remains unchanged at 500 Pa. So my conclusions based on this data is not that Desmin aggregation increases surface tension, rather that the mechanics of aggregates of cells expressing D399Y do not change with HS. This can also be seen on Figure S9. We would be curious to know what HS does to A21V aggregates. This may allow the authors to determine if some of the reported effects are due to overexpression of desmin.

4) The authors have collected a significant amount of immunostaining data to link aggregate mechanics to single cell changes. In the current presentation of the data, it is difficult to really assess their conclusions because, even with a very high zoom on the figures, individual cells are difficult to distinguish. For example, we could not judge whether we agreed or not with the changes in angle of contact reported in Figure 3. Similarly on S6 and S7, even with the zoomed images, it is very difficult to really distinguish individual cells, assess their morphological changes, and determine if there are any changes in cadherin, F-actin, etc. Furthermore, it would be useful to show the separate channels for those zooms so that the readers can see the changes in adhesion and F-actin. S8 is better in that respect.

5) To determine the effect of intercellular adhesion, the authors treat the aggregates with EGTA. This is not a very clean treatment because EGTA chelates extracellular calcium and can also deplete intracellular calcium if cells are left in calcium free medium for too long. A decrease in intracellular calcium concentration can affect myosin contractility. Therefore the effects of the EGTA treatment may be a combination of decrease in adhesion and myosin contraction.

6) Determination of the angle between cells, alpha:

We were surprised to not find any real image example of how alpha was determined. This point is however crucial, as the determination of this parameter is essential to disentangle cortical tension from effective adhesion. More specifically:

– The schematic of Figure 3c is not enough in our view to see how these measurements are done, with which precision, etc. Could the authors show some sample pictures? Can the authors better explain the image analysis procedure with FIJI: is this automatic, by hand? What is the precision of the measurement, and how does it compare to the variability observed in Figure 3d?

– Crysectioning can deform the object, which can lead to errors in alpha determination. Can the authors comment? Why not staining the spheroid directly and performing 3D imaging, since the estimation is only at the surface?

– Even though there must be a great number of cells analyzed (how many?), N = 2 spheroids may be too limited to reach the conclusions. Can the authors perform statistical tests for the data presented in Figure 3e and 3f?

7) Effect of heat shock and conclusions on desmin:

It is hard for us, at the moment, to draw conclusions on the desmin experiments. In particular, we find it odd that there were no discussion on the sole effect of heat shock (HS). Besides, we were surprised to not find results on how the A21V control cells react to the heat shock. More specifically:

– Can the authors plot on the same graph the data together? It would make it easier to compare them. Could they perform statistical analysis on the different pairs, when plotted together?

– It seems that the most striking result is that both surface tension and elasticity of desWT cells decrease under HS, while the values for desD399Y seem to not change. Can the authors comment on this? It seems that HS alone does a lot to the cells. How does the A21V control compare here, in particular?

– We believe that the authors would have the means to perform the same kind of analysis they did for Figure 3. It would be very interesting to measure, in every condition (A21V, desWT and desD399Y) how cortical tension and effective adhesion are affected by HS and by the desmin anti-sense mutation.

---

## [Author Response]

Essential revisions:1) The determination of γ and the Young's modulus relies on analysis of the aggregate shape. What are the assumptions on the structure of the aggregate and are these fulfilled? It looks like aggregate mechanics are dominated by the first few rows of cells at the surface. What assumptions are made about the aggregate interior? In many images like Figures3a, 4e, S2b, S6 or S7, it looks like the cells do not uniformly fill the aggregate interior and that there are clear gaps between cells. Is this an artefact of the fixation and cryosectioning? Or do the cells in the interior not form a cohesive material? If the cells do not form a cohesive interior, this may explain why the surface tension and the apparent elasticity of the aggregates are so tightly coupled (Figure S5). It would be useful to image some aggregates with live confocal microscopy or light sheet microscopy.

First we acknowledge the reviewers for their question regarding the actual inner structure of the aggregate. Confocal and 2-photon microscopy have been used to image fixed and immuno-labelled aggregates at 150 µm depth which is the maximal penetration depth for this kind of highly diffusive and absorbing structures (mainly related to nanoparticles whose degradation cannot be achieved without damaging the cells ).

The determination of the surface tension and of the Young’s modulus relies on a fluid continuous model meaning that multicellular aggregates can be approximated as a continuous medium. Our multicellular aggregates fulfilled the hypothesis of the fluid approximation as we have more than 10^5^ cells per aggregates and as they form a cohesive structure regarding the confocal images of the whole aggregate that we add to the figure 1 of the manuscript.

Using either confocal or two-photon microscopy, we imaged the multicellular aggregates over 150 µm without finding any hole in the structure as now emphasized in the manuscript.

Results page 3.

“Assuming that a multicellular cohesive aggregate can be modelled as a continuous elastic medium (as supported by confocal imaging, Figure 1b), surface tension and magnetic forces in volume compete to determine the equilibrium shape [14] while Young's modulus is at stake for the contact area [32] (Figure 1c and Figure 1 Supplement 3).”

Gaps sometimes observed in cryosections are thus artefacts due to fixation and cryosectionning processes. We added some extra 3D imaging obtained by confocal on figure 1 to prove this. Cryo-sections still allow to look at inner sections in the aggregate but frozing may introduce some artefacts due to local micro-cracks of the structure.

The coupling observed between surface tension and the apparent elasticity of the aggregates gives a characteristic length around 130 µm which is not modified by inhibitors, heat shock or desmin aggregation while actin network, contractility or cohesion properties are, thus precluding any role played by these components.

2) The link between single cell properties and aggregate mechanics could be strengthened. The mechanical measurements are done on the aggregates and the single cell mechanical changes are mostly inferred from γ using the angle of contact. So there no independent determinations of Tcc or Tcm. The changes in cytoskeletal organisation remain very qualitative and could be exploited by systematically quantifying the image data to show changes in F-actin, cadherin, and phosphomyosin. For example, the authors claim that desmin aggregation increases surface tension and cortical tension. They could determine if desmin aggregation leads to an increase in phosphomyosin in those locations. The methods indicate that phosphomyosin staining has been done but this appears nowhere in the figures – yet this is likely an important piece of data. On page 5, the authors suggest that the low change in effective adhesion in response to latA and EGTA addition may be due to an increase in adhesion. This could be verified and quantified from the immunostainings.

We would like to thank the reviewers for their suggestions which definitely increased the impact of the manuscript and the quality of the results. We add some images and comments in the manuscript to describe actin and phosphomyosin network in the different conditions. We focus on striking representative images that illustrate the organization of actin network and myosin activity.

First when looking over the usual suspects of cell mechanics which are actin network or contractility and adhesion, we determine the effects of inhibitors on actin networks and multicellular organization. For example we proved that blebbistatin action focus on impairing myosin activity not hindering adhesion or actin network (Figure 2c-d) while latrunculin A and EGTA have broader effects impacting both actin organization and cell-cell adhesion (Figure 3f). We add these comments in the manuscript referencing to confocal images of 3D aggregates obtained by confocal microscopy.

Results page 5

“In the multicellular aggregates environment, [blebbistatin] impairs the activity of myosins disrupting active phospho-myosin organization along actin filaments (Figure 2c and d).”

Results page 5

“In multicellular aggregates EGTA has an impact on the acto-myosin network (Figure 3f and Figure 2 supplement 1). By reducing cell-cell contacts, it impairs the formation of the contractile branched network of acto-myosin at the periphery of the aggregate. At the single cell level, F-actin becomes mainly cortical whatever the cell location inside the aggregate. It may be related to a lack of strong enough cell-cell adhesions to maintain actin branched network and to a co-regulation of actin and cadherin tension.”

And Results page 5

“Latrunculin A disrupts the actin filaments by binding to actin monomers thus precluding its polymerization [43], its addition gives some non-connected patches of actin filaments with a lack of long-range organization (Figure 3f and Figure 2 supplement 1).”

Second looking at a cellular model of desminopathies, we check that heat shock does not impair the actin network, myosin activity and cell organization while comparatively, desmin aggregation actually enhances myosin activity. These results are supported by actin and phosphomyosin immunolabelling and imaging in Figure 5. However, desmin aggregation points and myosin activity sites are not correlated. The presence of desmin aggregates seems to induce a large-scale reorganisation of myosin activity without modifying the actin network.

3) The main conclusions are largely confirmatory except for the role of desmin. The effects of decreased myosin contractility, depolymerisation of F-actin, and decreasing intercellular adhesion are largely expected. We find the data about Desmin's role confusing. In particular, the authors do not compare the mechanical properties of aggregates pre vs post HS. Yet, these comparisons seem to go against their main conclusion that desmin aggregation increases surface tension. In Figure 4f, we can see that prior to heat shock, γ is about 75mN/m in the two desmin lines and E is about 500-600Pa. After heat shock, the surface tension of the WT desmin decreases to 40-50 mN/m but the D399Y (that shows increases aggregation) does not change compared to no HS. Similarly, after HS, E for the WT desmin decreases to 300 Pa while for the D399Y it remains unchanged at 500 Pa. So my conclusions based on this data is not that Desmin aggregation increases surface tension, rather that the mechanics of aggregates of cells expressing D399Y do not change with HS. This can also be seen on Figure S9. We would be curious to know what HS does to A21V aggregates. This may allow the authors to determine if some of the reported effects are due to overexpression of desmin.

We acknowledge the authors for suggesting to describe carefully the effects of heat shock on cells while we were only focused on the description of differences between cells overexpressing either wild type or mutated desmin to highlight the impact of mutated desmin on tissue mechanical properties. In the present version of the manuscript, we use the A21V cells as control cells to determine the effect of heat shock on tissue organization, actin network, myosin activity and mechanical properties such as surface tension and Young's modulus. We clearly establish that A21V cells and cells overexpressing wild type desmin (desWT-Cl29) have the same behavior when submitted to heat shock. Heat shock reduces both surface tension and Young’s modulus. Conversely, desmin aggregation induced in cells overexpressing mutated desmin actually prevents these decreases while enhancing phosphomyosin activity.

Results are now presented in the modified Figure 4f-g and the new Figure 5 and are clearly explained in the results part.

“In this cellular model, desmin aggregation is induced by heat shock in 2D culture (Figure 4a-d and Figure 4 supplement 1) but also in multicellular spheroids (Figure 4e and Figure 4 supplement 2). In desmin mutated cells (desD399Y-Cl26), heat shock duration monitors the percentage of cells presenting desmin-aggregates with up to 30% of cells for 2 hours stress compared to a 2% ratio in the case of wild-type desmin expression (desWTCl29) (Figure 4c-d). As desmin aggregation is induced by heat shock in this model, the effect of protein aggregation and physical inducer have to be carefully deciphered. Measurements on A21V control cells allow the effect of a heat shock on surface tension and Young’s modulus to be verified. Heat shock decreases both Young’s modulus and surface tension of A21V control aggregates (Figure 4g and Figure 4 supplement 3). Conversely, heat shock does not impact roughness and local contact angles between cells at the periphery of the A21V multicellular aggregates (Figure 5a-b). The decrease of surface tension therefore translates at the cell tension level and both cell-medium and cell-cell tensions decrease (Figure 5e-f). This result is corroborated by the observation of actin network at the periphery of the multicellular aggregates: heat shock does not drastically modified actin distribution but slightly decreases the number of stress fibers (Figure 5c). The effect of heat shock on the properties of aggregates of cells overexpressing WT desmin (desWT- Cl29) is similar to that observed for A21V control cells (Figure 4g), but different for aggregates of cells overexpressing mutated desmin (desD399Y-Cl26). Focusing on desmin organization, we compare cells expressing mutated desmin (desD399Y-Cl26) to cells expressing wild-type desmin (desWT-Cl29). an increase in surface tension and elasticity dependent on the heat shock duration is observed in cells overexpressing mutated desmin (Figure 4f-g): the surface tension of spheroids of desD399Y-Cl26 cells is 1.8-fold higher than the one of desWT-Cl29 cells and their elasticity is 2-fold higher for 30 min heat shock. The effect is even greater for a 2 h heat shock with a 2 times higher surface tension of multicellular aggregates and a 3 times higher Young’s modulus for cells expressing mutated desmin compared to cells expressing the wild-type. The effects on mechanical properties are therefore correlated with the percentage of cells containing desmin aggregates.

Looking at the local geometry at the surface of multicellular aggregates, cells overexpressing desmin also show a predominance of tension at the cell medium interface (Figure 5). When desmin overexpression is induced, cortical tension at the cell-medium interface slightly increases (by a factor of 2) comparing desWT-Cl29 and desD399Y-Cl26 spheroids while effective adhesion is less impacted. Overexpression of wild-type and mutated desmin does not change phosphomyosin distribution on the overall aggregate (Figure 5 supplement 1) and desmin aggregates are not colocalized with contractile points (Figure 5 and Figure 5 supplement 1). However mutated desmin aggregation strengthens long range phospho-myosin filaments at the periphery of the spheroid (Figure 5e) which increases acto-myosin network contractility.”

4) The authors have collected a significant amount of immunostaining data to link aggregate mechanics to single cell changes. In the current presentation of the data, it is difficult to really assess their conclusions because, even with a very high zoom on the figures, individual cells are difficult to distinguish. For example, we could not judge whether we agreed or not with the changes in angle of contact reported in Figure 3. Similarly on S6 and S7, even with the zoomed images, it is very difficult to really distinguish individual cells, assess their morphological changes, and determine if there are any changes in cadherin, F-actin, etc. Furthermore, it would be useful to show the separate channels for those zooms so that the readers can see the changes in adhesion and F-actin. S8 is better in that respect.

We apologize for the poor quality of the images due to compression process of the.pdf file that has to be limited. We increase image resolution and add zoomed images as well as separate channels data in all figures.

5) To determine the effect of intercellular adhesion, the authors treat the aggregates with EGTA. This is not a very clean treatment because EGTA chelates extracellular calcium and can also deplete intracellular calcium if cells are left in calcium free medium for too long. A decrease in intracellular calcium concentration can affect myosin contractility. Therefore the effects of the EGTA treatment may be a combination of decrease in adhesion and myosin contraction.

We thank the reviewers for their relevant comment and we have now added some additional data to support the idea that EGTA actually impairs both cell-cell adhesion and the actin network by rounding up cells. Conversely total active myosin concentration is not modified by EGTA addition and phosphomyosin colocalizes with cortical actin thus precluding any direct action of EGTA on contractility. However as actin becomes mainly cortical due to a swollen of cell-cell junctions in EGTA treated spheroids, cell tension may be hindered. Subfigure 3f has now been added to give the distribution of both actin and phosphomyosin while Figure 2 supplement 1 has now enlarged to better visualise effects of EGTA. A comment has also been added to the manuscript. It is important to note that EGTA has been used because of its modulated action as we still need to form a cohesive spheroid.

“EGTA as a calcium chelator, reduces the efficiency of this homophilic adhesion and modulates cell-cell adhesion strength. In multicellular aggregates EGTA has an impact on the acto-myosin network (Figure 3f and Figure 2 supplement 1). By reducing cell-cell contacts, it impairs the formation of the contractile branched network of acto-myosin at the periphery of the aggregate. At the single cell level, F-actin becomes mainly cortical whatever the cell location inside the aggregate. It may be related to a lack of strong enough cell-cell adhesions to maintain actin branched network and to a co-regulation of actin and cadherin tension.”

6) Determination of the angle between cells, alpha:We were surprised to not find any real image example of how alpha was determined. This point is however crucial, as the determination of this parameter is essential to disentangle cortical tension from effective adhesion. More specifically:– The schematic of Figure 3c is not enough in our view to see how these measurements are done, with which precision, etc. Could the authors show some sample pictures? Can the authors better explain the image analysis procedure with FIJI: is this automatic, by hand? What is the precision of the measurement, and how does it compare to the variability observed in Figure 3d?– Crysectioning can deform the object, which can lead to errors in alpha determination. Can the authors comment? Why not staining the spheroid directly and performing 3D imaging, since the estimation is only at the surface?– Even though there must be a great number of cells analyzed (how many?), N = 2 spheroids may be too limited to reach the conclusions. Can the authors perform statistical tests for the data presented in Figure 3e and 3f?

We agree with reviewers that the addition of high resolution images of cells at the interface helps to define the local angle. As suggested by reviewers we now add examples in Subfigure 3d.

Enhanced quality zooms of the interface should also be helpful to better visualize tissue roughness and change in cellular behavior.

To measure local angle, image analysis is performed by hand using FIJI. FIJI's preprogrammed analysis tools failed to analyze our data. Particular care is taken to define the contact line between two cells and then to determine the local cell contours. The results of two independent operators are compared for each sample. Accuracy of the measurements are estimated from this comparison. It is less than 5° which is very low compared to the variability of the samples (usually more than 15°). We now increase the number of independent experiments to extract the local angle. Each point in the graphs represent a different cell.

As suggested by reviewers, we also look at contact angles using confocal microscopy on fixed and immuno-labelled spheroids. We compared the results and found no significant difference between the two preparation protocols as emphasized in Figure 3 supplement 1.

7) Effect of heat shock and conclusions on desmin:It is hard for us, at the moment, to draw conclusions on the desmin experiments. In particular, we find it odd that there were no discussion on the sole effect of heat shock (HS). Besides, we were surprised to not find results on how the A21V control cells react to the heat shock. More specifically:– Can the authors plot on the same graph the data together? It would make it easier to compare them. Could they perform statistical analysis on the different pairs, when plotted together?– It seems that the most striking result is that both surface tension and elasticity of desWT cells decrease under HS, while the values for desD399Y seem to not change. Can the authors comment on this? It seems that HS alone does a lot to the cells. How does the A21V control compare here, in particular?– We believe that the authors would have the means to perform the same kind of analysis they did for Figure 3. It would be very interesting to measure, in every condition (A21V, desWT and desD399Y) how cortical tension and effective adhesion are affected by HS and by the desmin anti-sense mutation.

We acknowledge the authors for suggesting to describe carefully the effects of heat shock on cells while we were rather focused on the description of differences between cells overexpressing either wild type or mutated desmin to highlight the impact of mutated desmin on tissue mechanical properties. In the present version of the manuscript, we use the A21V cells as control cells to determine the effect of heat shock on tissue organisation, actin network, myosin activity and mechanical properties such as surface tension and Young's modulus. We clearly establish that A21V cells and cells overexpressing wild type desmin (desWT-Cl29) have the same behavior when submitted to heat shock. Heat shock reduces both surface tension and Young’s modulus. These results are presented both in Figure 5 where all the data are plotted together and in the dedicated Figure 4 supplement 3 to clearly looked at changes due to heat shock.

Surprisingly, this change in surface tension and elastic modulus is not accompanied by a change in actin structure at the interface or a decrease in acto-myosin contractility, suggesting instead an action on membrane components that we failed to identified.

Comparatively, cells expressing mutated desmin have a more intense and well organized myosin activity over actin filaments meaning that desmin aggregation induces a change in acto-myosin contractility. However no colocalisation is observed between protein aggregations sites and phosphomyosin distribution.